# Chitosan Nanoparticle-Mediated Delivery of Curcumin Suppresses Tumor Growth in Breast Cancer

**DOI:** 10.3390/nano14151294

**Published:** 2024-07-31

**Authors:** Barnalee Mishra, Amit Singh Yadav, Diksha Malhotra, Tandrima Mitra, Simran Sinsinwar, N. N. V. Radharani, Saroj Ranjan Sahoo, Srinivas Patnaik, Gopal C. Kundu

**Affiliations:** 1School of Biotechnology, KIIT Deemed to be University, Bhubaneswar 751024, India; barnaleemishra12@gmail.com (B.M.); yadav.singh.amit@gmail.com (A.S.Y.); diksha.ksbt@gmail.com (D.M.); tandrima.mitra@gmail.com (T.M.); simransinsinwar13@gmail.com (S.S.); nnvradha@gmail.com (N.N.V.R.); srinivas.patnaik@kiitbiotech.ac.in (S.P.); 2National Centre for Cell Science (NCCS), Pune 411007, India; 3Kalinga Institute of Medical Sciences (KIMS), Bhubaneswar 751024, India; saroj.sahoo1@kims.ac.in

**Keywords:** curcumin, breast cancer, chitosan, nanoparticle, drug delivery, cancer therapy

## Abstract

Curcumin is a nutraceutical known to have numerous medicinal effects including anticancer activity. However, due to its poor water solubility and bioavailability, the therapeutic impact of curcumin against cancer, including breast cancer, has been constrained. Encapsulating curcumin into chitosan nanoparticles (CHNPs) is an effective method to increase its bioavailability as well as antitumorigenic activity. In the current study, the effects of curcumin-encapsulated CHNPs (Cur-CHNPs) on cell migration, targeted homing and tumor growth were examined using in vitro and in vivo breast cancer models. Cur-CHNPs possessed a monodispersed nature with long-term colloidal stability, and demonstrated significant inhibition of cell viability in vitro, which was potentiated by 5-Fluorouracil (5-FU). Outcomes of the in vivo imaging studies confirmed effective tumor targeting and retention ability of Cur-CHNPs, thereby suppressing breast tumor growth in mice models. Overall, the results demonstrated that Cur-CHNPs could be an effective candidate drug formulation for management of breast cancer.

## 1. Introduction

Cancer is among the most common causes of mortality worldwide, accounting for approximately 9.7 million cancer-related deaths in 2022 [1]. According to GLOBOCAN 2022, among females, breast cancer is the most commonly identified cancer (11.6%) worldwide [1]. Furthermore, it is a prominent and persistent cause of mortality among women [1]. Eradication of tumors by surgery followed by radiation and chemotherapy are conventional methods of treatment for breast cancer [2]. The use of naturally occurring therapeutic agents and nutraceuticals in cancer treatment has been widely investigated as these products may have potent antitumor effects with significantly lower side effects [3]. Curcumin, a polyphenolic nutraceutical, is derived from *Curcuma longa* (the rhizome of turmeric) and has diverse pharmacological properties, including antioxidant and anticancer properties [4]. Specifically, curcumin has been shown to exert an antitumor effect by inhibiting cell growth and tumor angiogenesis and inducing apoptosis [5]. Despite these therapeutic activities, curcumin exhibits poor solubility, stability and bioavailability, limiting its efficacy [6].

Nanotechnology provides suitable tools that can overcome the drawbacks of curcumin and deliver the drug to its suitable target site, leading to enhanced efficacy of the drug [7]. Among various types of nanoparticles, naturally occurring polymer-based nanocarriers such as chitosan are more efficient owing to their biocompatibility, low cost and environmental friendly nature [8]. Chitosan nanoparticles (CHNPs) are extensively studied nanocarriers in various medical fields, including cancer therapy, and have obtained recognition for the easy administration of the carrier drug into host cells via oral and intravenous methods [8]. Chitosan forms a complex via inter- and intra-crosslinks with polyanions such as tripolyphosphate (TPP); entrapment of bioactive compounds can be performed by the ionic gelation method [9]. Moreover, CHNPs can be easily surface modified with specific ligands for targeted drug delivery to selective tissues [10]. Yadav et al. demonstrated that peptide ligand-conjugated CHNPs selectively delivered raloxifene to cancer cells in a targeted and efficient manner, leading to the suppression of breast tumor growth, migration and angiogenesis [11]. Several studies have shown ionically crosslinked CHNPs to be efficient nanocarriers for various drugs, including curcumin, as cancer therapeutics [11,12,13,14,15,16,17,18,19]. However, studies regarding curcumin-encapsulated CHNPs (Cur-CHNPs) have been restricted to physicochemical characterization and the assessment of in vitro efficacy [15,16]. To establish Cur-CHNPs as an anticancer therapeutic, exclusive preclinical assessment of biodistribution, therapeutic efficacy, biocompatibility and nonspecific toxicities, the present study is aimed to prepare a stable Cur-CHNPs formulation and to perform a detailed examination of the cytotoxicity, tumor accumulation and retention, antitumor efficacy and acute or subchronic toxicities of the formulation in the presence or absence of 5-Fluorouracil (5-FU) using in vitro and in vivo breast cancer models.

## 2. Materials and Methods

### 2.1. Culturing of the Cell Line and Primary Culture

MDA-MB-231 cells (human breast carcinoma cells) were cultured in DMEM (Gibco; Thermo Fisher Scientific, Inc., Waltham, MA, USA), comprising 10% FBS (Gibco; Thermo Fisher Scientific, Inc.), 100 units of penicillin and 100 μg/mL of streptomycin, in an incubator supplied with 5% CO_2_.

Breast cancer clinical specimens were used to establish the primary cell cultures. The breast cancer clinical specimens were collected from the Kalinga Institute of Medical Sciences (KIMS), KIIT University, India, which was approved by the Institutional Ethics Committee of KIMS with patient-informed consent. These tissues specimens were washed with antibiotic and DMEM media, then cut into small pieces using tumor dissociation solution (Miltenyi Biotec GmbH, Macquarie Park, Australia) and were incubated (37 °C) for 30 min. Then, the suspension of cells was centrifuged. The supernatant was discarded and pellets were resuspended in DMEM and passed through a 40 µm cell strainer. The cells were then seeded into 12-well plates, incubated and maintained in DMEM comprising 10% FBS, 100 units of penicillin, 100 μg/mL streptomycin, 10 ng/mL insulin-like GF (IGF) and 10 ng/mL epidermal growth factor (EGF).

### 2.2. Preparation of Nanoparticles

CHNPs were prepared using the ionic gelation method with minor modifications [9]. For this, 10 mg chitosan was added to 10 mL 1% acetic acid. The pH was adjusted to 5.0 with 1 M NaOH. To the chitosan solution, 1 mg/mL TPP was added dropwise at a ratio of 2.5:1 (chitosan: TPP) and CHNPs were formed. For the synthesis of Cur-CHNPs, 25 mg curcumin was dissolved in 1 mL dimethyl sulfoxide, and then a defined volume of curcumin solution ranging from 2.5 to 50% (weight % of chitosan) was mixed into the chitosan solution preceding the addition of TPP.

### 2.3. Size and Zeta Potential Characterization

The size of the CHNPs and their zeta potential were analyzed by DLS (dynamic light scattering: Zetasizer Nano ZS instrument; Malvern Instruments Ltd., Malvern, UK). To evaluate the long-term colloidal stability of Cur-CHNPs, the particle size was observed at day 0, 7, 14, 21, 42 and 74. Each sample was measured in triplicate.

### 2.4. UV-Visible Spectroscopy

The UV-visible absorption spectra of free curcumin, CHNPs and Cur-CHNPs were recorded at 200–600 nm in quartz cuvettes, with a UV-vis spectrophotometer (Cary 100 UV-Vis; Agilent Technologies, Inc., Santa Clara, CA, USA).

### 2.5. Fourier Transform Infrared (FTIR) Spectroscopy

The FTIR spectra of CHNPs, curcumin and Cur-CHNPs were verified using an infrared spectrophotometer. In brief, 5 mg of each sample was mixed with potassium bromide. The spectra were then measured with ±4 cm^−1^ resolution, over a range from 400 to 4000 cm^−1^. FTIR curves were generated and analyzed using Essential FTIR Infrared File Viewer (v 1.50.282).

### 2.6. Encapsulation Efficiency (EE)

To determine the EE of Cur-CHNPs, the Cur-CHNPs suspension was first centrifuged at 12,000 rpm for 30 min. The amount of curcumin in the supernatant was then measured using UV-visible spectrophotometry at 425 nm. The formula used for the calculation of percentage of EE is as follows:EE% = [(total drug − amount of drug in supernatant)/total drug] × 100.

### 2.7. In Vitro Cellular Uptake

The in vitro cellular uptake of CHNPs by MDA-MB-231 cells was evaluated using confocal microscopy as described previously [20]. Cy5.5-NHS was dissolved in dimethyl sulfoxide and added dropwise to the chitosan solution (1 mg/mL). The mixture was then incubated for 12 h, after which free Cy5.5 was removed. TPP was added to the solution to form CHNPs conjugated with Cy5.5 (Cy5.5-CHNPs) through electrostatic interactions. MDA-MB-231 cells were then incubated with Cy5.5-CHNPs (50 and 100 µg/mL) for a period of 2 h. Images of the cells were collected by confocal microscopy (Leica Microsystems GmbH, Wetzlar, Germany) and the data were analyzed using LAS AF Lite software (Version 3.3).

### 2.8. Wound-Healing Assay

MDA-MB-231 cells were cultured in a 12-well plate. Using a sterile tip, wounds were created in the monolayered cells. Treatment was performed with free CHNPs, curcumin, Cur-CHNPs, 5-FU and Cur-CHNPs along with 5-FU (10 µM of equivalent concentration of curcumin) for 12 h. After incubation, images were collected using a phase contrast microscope (Nikon Corporation, Tokyo, Japan) with three high powered fields (HPFs). The area of migration was analyzed using Image J (1.46r software, NIH image analysis software) and presented as a bar graph.

### 2.9. Western Blotting

MDA-MB-231 cells were treated for 24 h with CHNPs, curcumin, Cur-CHNPs, 5-FU and Cur-CHNPs plus 5-FU (10 μM). The cells were lysed, the lysates containing total proteins were resolved through SDS-PAGE and subsequently analyzed by Western blot. The levels of various proteins were detected by incubating with their respective primary antibodies such as PI3K (Cat No. P851637, Santa Cruz Biotechnology, Inc., Dallas, TX, USA; 1:1000 dilution), p-Akt (Cat No. 271966, Santa Cruz Biotechnology, Inc.; 1:1000 dilution) and the proangiogenic proteins, osteopontin (OPN) (Cat No. 73631, Santa Cruz Biotechnology, Inc.; 1:1000 dilution) and vascular endothelial growth factor (VEGF) (Cat No. 7269, Santa Cruz Biotechnology, Inc.; 1:1000 dilution) and followed by incubation with secondary (HRP conjugated) antibodies. Actin was used as the loading control.

### 2.10. Antitumor Efficacy of Cur-CHNPs

The antitumor efficacy of Cur-CHNPs alone or along with 5-FU was evaluated in vitro using MDA-MB-231 cells and breast cancer clinical specimen-derived primary cells using the MTT assay. For the IC_50_ estimation, MDA-MB-231 cells were seeded into a 96-well plate (1 × 10^4^ cells per well) and then treated with various concentrations of free curcumin (0, 5, 10, 15, 20, 25 and 50 µM) at 37 °C for 24 h. In distinct experiments, MDA-MB-231 cells and primary cultured cells (1 × 10^4^ cells per well) were treated with free curcumin (10 µM), Cur-CHNPs (10 µM) alone or along with 5-FU (10 µM) or 5-FU (10 µM) alone at 37 °C for 24 h. MTT was added to the cells and incubated for 4 h, after which resultant formazan crystals was dissolved in isopropanol. The optical density of the solution was assessed using a plate reader at 570 nm. The percentage cell viability was determined with respect to the control cells and the IC_50_ was estimated using GraphPad Prism 5.0 software (Dotmatics, Bishop’s Stortford, UK).

### 2.11. Biodistribution and Tumor Homing of CHNPs Using NOD/SCID Mice Model

The tumor targeting ability of CHNPs was examined via in vivo biodistribution in tumor-bearing NOD/SCID mice by using an IVIS Spectrum (Xenogen Corporation, Alameda, CA, USA). For this analysis, MDA-MB-231 cells (2 × 10^5^) were injected into the right mammary fat pad of 6-week-old NOD/SCID mice. After the development of tumors, mice were fed Cy5.5-CHNPs and images of NIRF were captured at different time intervals (0–72 h). At the specific time points (24, 48 and 72 h), the tumors and different major organs were removed and images were captured. The NIRF intensities in the regions of interest were calculated and presented as bar graphs.

### 2.12. In Vivo Antitumor Efficacy Studies in Orthotopic Breast Cancer NOD/SCID Mice Models

To evaluate the in vivo anticancer efficacy of Cur-CHNPs, MDA-MB-231 cells (2 × 10^5^) were injected into mammary fat pads of NOD/SCID mice and tumors were developed [12]. Randomly, the mice were distributed into four groups (3 mice per group) and free curcumin, CHNPs or Cur-CHNPs (10 mg/kg body weight) were administered orally every alternate day for 2 weeks. Untreated mice were used as the positive control. Once the tumors reached a volume of ~2000 mm^3^ and diameter of >20 mm, mice were euthanized using CO_2_ asphyxiation by maintaining the CO_2_ flow rate at 30–70% volume per min. Death was confirmed by detecting the cessation of heartbeat, as well as areflexia, as per the IACUC guidelines. The tumor size was measured, and the volume was calculated using the following formula:Tumor volume = π/6(d1 × d2)^3/2^
where d1 is length of the tumor and d2 is width of the tumor. The tumors were harvested on the 16th day of the experiments, and the mean tumor volume and weight were calculated and statistically analyzed.

### 2.13. Assessment of the Acute and Subchronic Toxicities of Cur-CHNPs in BALB/c Mice

To assess the acute toxicity of Cur-CHNPs, male BALB/c mice (4–6 weeks old) were randomly treated orally with either free curcumin or Cur-CHNPs (20 mg/kg of body weight) for 7 successive days. Untreated mice were used as the control. Blood samples were collected from the surviving animals from all groups (4 mice per group) on day 7 for clinicopathological evaluation. The blood samples were collected by puncturing the orbital plexus using a fine capillary tube under light isoflurane anesthesia (2–3% isoflurane inhalation). The mice were then euthanized by CO_2_ asphyxiation following the blood collection.

In total, ~0.3 mL of blood was taken in vials containing 4% EDTA and used for hematological analysis. A drop of blood was transferred onto a clean glass slide and a thin smear was stained with Leishman’s stain. Then, leukocyte counting was performed manually by measuring 100 leukocytes for each animal under a light microscope. Furthermore, various hematological parameters of the blood samples were analyzed using a Beckman Coulter automated veterinary hematology analyzer.

Furthermore, ~1.0 mL of blood was collected from each mouse in separate centrifuge tubes in order to separate the serum. The blood was allowed to coagulate at room temperature and centrifugation was performed at 2500–3000 rpm for 10 min to separate the serum. The separated serum was used for assessment of the biochemical parameters associated with liver and kidney functions using an Elitech Selectra (ELITechgroup, Puteaux, France) automated clinical chemistry analyzer.

To assess subchronic toxicity, both male and female BALB/c mice were treated with either free curcumin or Cur-CHNPs (10 mg/kg body weight dose) each alternate day for 28 days. Blood samples were collected from the surviving animals from all groups (control, curcumin and Cur-CHNP groups; 5 mice per group for each sex) on day 28 for clinicopathological evaluation. The assessment of hematological and biochemical parameters was performed as aforementioned.

### 2.14. Statistical Analysis

Data are represented as the mean ± SEM from three independent experiments. GraphPad Prism 5.0 software was used to analyze the data using a one-way ANOVA or an unpaired Student’s *t*-test.

## 3. Results

### 3.1. Physicochemical Characterization of Nanoparticles

CHNPs were prepared by the ionic gelation technique, which results in the noncovalent interaction between the chitosan (positively charged) and polyanions, especially TPP (negatively charged), which act as crosslinker. The size distribution and zeta potential of the nanosuspension were characterized using the DLS technique. In the present study, the size of optimized CHNPs alone was 85 nm with polydispersity index (PDI) 0.232 and zeta potential of +26 mV (Figure 1A,B), whereas the size of optimized Cur-CHNPs was found to be 118 nm (PDI, 0.255) with a zeta potential of +14.2 mV (Figure 1C,D). The increase in size of Cur-CHNPs compared with CHNPs alone is one of the indicators of a successful encapsulation. The solubilities of Cur-CHNPs and free curcumin were analyzed in aqueous solution and in DMSO. Cur-CHNPs demonstrated a higher solubility in aqueous solution compared with free curcumin, but the reverse effect was observed in DMSO (Appendix A).

The long-term colloidal stability of the Cur-CHNP formulation at different time points was further examined and the results revealed that there were no substantial changes in size of the Cur-CHNPs even after 74 days (Figure 1E). This indicated that the Cur-CHNPs did not aggregate or degrade and possessed high colloidal stability.

The optical properties of free curcumin, CHNPs and Cur-CHNPs were analyzed by UV-visible spectroscopy between 200 and 600 nm. The UV-visible spectrum of curcumin and the CHNPs had absorption peaks at 425 and 260 nm, respectively. Furthermore, UV-visible spectra of free curcumin demonstrated distinct peaks at 425 nm in the visible region, suggesting the presence of curcumin in the nanoformulation (Figure 1F).

To confirm the interaction and conversion of chitosan and TPP into CHNPs as well as the encapsulation of curcumin in Cur-CHNPs, FTIR was performed. In the chitosan FTIR spectra, peaks were observed at 3360 and 3290 cm^−1^, which corresponded to -OH and -NH_2_ stretching vibrations, respectively. Similarly, the occurrence of peaks at 1646 cm^−1^ and 1576 cm^−1^ referred to CONH_2_ (amide) and highly deacetylated amine (N-H bending) groups, respectively. The CHNP FTIR spectra demonstrated a shift in peaks attributed to the amide group (1646 cm^−1^ to 1630 cm^−1^) and the N-H bending region (1576 cm^−1^ to 1539 cm^−1^), indicating an interaction between the amine group of chitosan and TPP. Moreover, the CHNP FTIR spectra had an absorption peak at 1217 cm^−1^ (corresponding to P-O stretching) confirming the presence of phosphate residues in the nanoparticles. In addition, the peak at 3360 cm^−1^ (corresponding to -OH groups) observed in chitosan was wider and shifted to 3190 cm^−1^ in the CHNPs, indicating enhanced hydrogen bonding (Appendix A). The curcumin spectra had a broad peak in the range of 3500–4000 cm^−1^ due to the OH stretching of phenol. Similarly, a C=C stretching vibration at 1425 cm^−1^ and a peak at 1506 cm^−1^ were observed, which was accredited to diverse vibrations including stretching carbonyl bond vibrations ν(C=O) and in-plane bending vibrations about aromatic δ CC-H of keto and enol configurations, and stretching vibrations around aromatic ν C-C bonds of the keto and enolic forms of curcumin. The IR bands at 1023, 810 and 715 cm^−1^ could be assigned to ν (C-H) out-of-plane vibrations of the aromatic ring (Figure 2A). Compared with curcumin, a peak shift was observed and a new sharp peak appeared in the Cur-CHNPs spectra at 2875 cm^−1^. Compared with the CHNPs, new peaks at 1410, 698 and 650 cm^−1^ were observed in the Cur-CHNPs spectra, which may correspond to the curcumin IR peaks at 1425 and 715 cm^−1^ (Figure 2A). Shifts in these peaks indicated an interaction between chitosan and curcumin in these regions. Overall, these data confirmed the encapsulation of curcumin by CHNPs.

Curcumin encapsulation by CHNPs was optimized by adding the drug at different ratios with respect to chitosan while preparing the nanoparticles. As shown in Table 1, the EE increased from 63 to 94% as the amount of drug added increased from 2.5 to 50% (*w*/*w* of chitosan) (Appendix A). However, nanosized particles were only observed when 2.5–25% of the drug was added, while large drug aggregates were observed at 30 and 50% (Appendix A). Furthermore, UV-visible spectroscopic analysis of the Cur-CHNPs also demonstrated a steady increase in the curcumin peak size when the drug was increased from 2.5 to 25%, suggesting a higher curcumin encapsulation. However, in contrast to the encapsulation results, a decrease in the curcumin peak was observed at the 30 and 50% drug ratio (Figure 2B). This may be the result of undissolved curcumin precipitation at the 30 and 50% addition ratio, which would lead to a lower absorbance in the spectroscopic analysis, leading to overestimation of the EE. The Cur-CHNP formulation prepared with a 25% drug ratio was used as the lead formulation for all subsequent in vitro and in vivo studies.

### 3.2. In Vitro Cellular Uptake of CHNPs

Efficient cellular internalization of nanocarriers is necessary for potent therapeutic effects. Therefore, the cellular uptake of CHNPs by MDA-MB-231 cells was examined. The CHNPs were conjugated with the NIRF dye, Cy5.5, and then added to the cells at 50 and 100 µg/mL for 2 h. Images of the cells were collected using fluorescence microscopy. The data indicated a high, dose-dependent Cy5.5 fluorescence intensity in the MDA-MB-231 cells, suggesting efficient cellular uptake of the nanoparticles (Figure 3A).

### 3.3. Assessment of the In Vitro Antitumor Efficacy of Cur-CHNPs

To determine the effect of Cur-CHNPs on cell viability, MDA-MB-231 cells were cultured overnight in 96-well plates and treated with different concentrations of free curcumin (0–50 µM) at 37 °C for 24 h. The cytotoxicity of curcumin was subsequently analyzed by MTT assay. The IC_50_ of curcumin was estimated to be 21 µM (Appendix A).

In addition, MDA-MB-231 cells were treated with free curcumin (10 µM) or Cur-CHNPs (10 µM) in the absence or presence of 5-FU (10 µM) for 24 h, before the MTT assay was performed. The primary cell culture from breast cancer clinical specimens was established and analyzed by phase-contrast microscopy (Figure 3C). The results demonstrated that Cur-CHNPs remarkably decreased the cell viability of MDA-MB-231 and the primary culture, compared with free curcumin (Figure 3B and D, respectively). The treatment of curcumin and 5-FU in combination demonstrated a significant inhibition of the MDA-MB-231 and primary breast cancer cell viability; however, Cur-CHNPs in combination with 5-FU exhibited a much higher cytotoxicity compared with the free curcumin and 5-FU combination (Figure 3B,D). Furthermore, CHNPs alone did not demonstrate any significant cytotoxicity towards MDA-MB-231 cells (Figure 3B). These results suggested that the Cur-CHNPs possessed a higher antitumor efficacy and demonstrated a better effect when combined with 5-FU, compared with free curcumin.

### 3.4. Effect of Cur-CHNPs on Breast Cancer Cell Migration

To examine the effect of Cur-CHNPs in the absence or presence of 5-FU on breast cancer cell migration, a wound-scratch assay was performed. MDA-MB-231 cells were cultured as a monolayer and wounds of equal width were created. The cells were treated with CHNPs, free curcumin, Cur-CHNPs, 5-FU and Cur-CHNPs plus 5-FU for 0 and 12 h. After 12 h, images were captured and analyzed. The results demonstrated that Cur-CHNPs in combination with 5-FU attenuated the motility of MDA-MB-231 cells, compared with Cur-CHNPs or free curcumin (Figure 4A). The results were quantified and presented as a bar chart (Figure 4B). These results suggested the antimigratory and potentially antimetastatic properties of Cur-CHNPs and Cur-CHNPs plus 5-FU.

### 3.5. Effect of Cur-CHNPs on PI3K and Akt Activation and OPN and VEGF Expression

To study the effect of Cur-CHNPs in the absence or presence of 5-FU on PI3K and Akt activation and OPN and VEGF expression, MDA-MB-231 cells were treated with specific nanoformulation and Western blot analysis was performed. The results demonstrated that Cur-CHNPs in combination with 5-FU attenuated PI3K and Akt activation (Figure 4D). Moreover, Cur-CHNPs, either alone or in combinatorial treatment with 5-FU, suppressed the expression of OPN and VEGF (Figure 4C). These results suggested that Cur-CHNPs in combination with 5-FU inhibited PI3K and Akt activation and VEGF and OPN expression in the cells.

### 3.6. In Vivo Biodistribution and Tumor Homing of CHNPs

Prior to examining the antitumor efficacy of Cur-CHNPs, the biodistribution and in vivo tumor targeting ability of CHNPs were assessed using an In Vivo Imaging System (IVIS) Spectrum. After treatment with Cy5.5-CHNPs to the tumor-bearing mice, the NIRF images were captured at specific time points. The data revealed strong fluorescence in the abdominal region of the mice after 1 and 6 h of treatment (Figure 5A). However, after 24 h, most nanoparticles were absorbed and preferentially accumulated in tumor tissue (Figure 5A). A high NIRF signal was observed in the tumors compared with other organs, as shown by ex vivo imaging (Figure 5B). The NIRF intensity of Cy5.5-CHNPs was quantified for the region of interest in each organ and plotted as a bar graph (Figure 5C). The results suggested that the accumulation of nanoparticles was higher in the tumors compared with the other organs, demonstrating the high tumor targeting ability of CHNPs.

The time-dependent biodistribution of the CHNPs was subsequently monitored up to 72 h post-treatment. The NIRF images showed that the tumors maintained the maximum NIRF intensity up to 72 h (Figure 5D). The ex vivo images demonstrated a strong NIRF signal in the tumors compared with the other organs up to 72 h, which further validated the high tumor targeting ability of CHNPs (Figure 5E). Moreover, quantification of the NIRF signal demonstrated retention of the maximal intensity for up to 72 h, which indicated the high retention ability of CHNPs (Figure 5F). Therefore, these results suggested that CHNPs possessed effective tumor targeting and retention ability, making them a useful tool for enhanced curcumin delivery for breast cancer therapy.

### 3.7. In Vivo Antitumor Efficacy of CHNPs

The effect of Cur-CHNPs on NOD/SCID mice was further examined. Orthotopic breast tumor models were developed by injecting MDA-MB-231 cells into the mammary fat pads of NOD/SCID mice. The tumor growth was carefully monitored until the tumor volume reached ~2000 mm^3^. Then, the mice were treated with free curcumin or Cur-CHNPs (curcumin dose of 10 mg/kg of body weight) and the tumor growth was observed with respect to the untreated control. The results revealed that treatment with Cur-CHNPs significantly suppressed breast tumor growth, compared with the control and free curcumin treatment (Figure 6A). Furthermore, CHNPs alone did not show significant antitumor activity. The tumor volume and weight were also found to be significantly lower in the Cur-CHNPs-treated mice (Figure 6B,C). Overall, these results suggested that Cur-CHNPs had a significant in vivo antitumor efficacy as demonstrated by preclinical organotropic models.

### 3.8. In Vivo Toxicity Assessment

An acute and subchronic toxicity assessment of Cur-CHNPs in healthy BALB/c mice was also performed.

#### 3.8.1. Acute Toxicity

For the acute toxicity assessment, 4–6-week-old healthy male BALB/c mice were orally administered with free curcumin or Cur-CHNPs (20 mg/kg of body weight) for 7 consecutive days. Blood was drawn from the mice and subjected to hematological and biochemical analyses to assess hematoxicity alongside liver and kidney function. No significant alteration in hematological parameters was observed, even after 1 week of treatment. In addition, no significant differences were observed in the white blood cell and red blood cell counts of the Cur-CHNPs-treated mice compared with the control (Figure 7A(i,ii)). Moreover, the hemoglobin and mean corpuscular volume were also unchanged (Figure 7A(iii,iv)). No significant differences were observed in the number of platelets, neutrophils, lymphocytes and monocytes, suggesting that Cur-CHNP treatment did not alter immunogenicity (Figure 7A(v–viii)). 

In addition, to study the acute toxicity of Cur-CHNPs on the liver and kidneys, their effect on associated biochemical functions were investigated. No significant changes in alkaline phosphatase (ALP), alanine aminotransferase (ALT) and aspartate aminotransferase (AST) levels were observed in the treatment groups compared with the control, indicating that Cur-CHNPs did not cause cellular damage to the liver (Figure 7B(i–iii)). Alterations in these parameters are a potential risk factor of liver disease. Further results revealed no significant differences in the serum bilirubin and albumin levels in the control and treatment groups (Figure 7B(v,vi)). Serum levels of urea, creatinine and blood urea nitrogen (BUN) are common signatures of renal function, and enhanced concentrations of creatinine, urea and BUN are possible causes of kidney damage and renal failure. No changes were observed in these parameters in the treatment groups compared with the control, suggesting that Cur-CHNPs did not incur renal toxicity in mice (Figure 7B(iv,vii,viii)). Overall, these results suggested that Cur-CHNPs did not cause any acute toxicities in mice.

#### 3.8.2. Subchronic Toxicity

The subchronic toxicity of Cur-CHNPs in BALB/c mice was also examined. Both male and female BALB/c mice were orally administered with either free curcumin or Cur-CHNPs (both at 10 mg/kg of body weight) on every alternate day for 28 days. Blood samples were collected from all the surviving animals from all treatment groups (5 mice per group for each sex) for clinicopathological examination. Similar to the acute toxicity assessment results, no significant changes were observed in both the hematological and biochemical parameters of the Cur-CHNPs-treated male BALB/c mice, even after 28 days (Appendix A). In addition, no apparent subchronic toxicities were observed in female BALB/c mice, as indicated by no significant changes in hematological and biochemical parameters following Cur-CHNPs treatment for 28 days (Figure 8A,B). Therefore, these observations demonstrated that Cur-CHNPs did not cause acute or subchronic toxicity using in vivo models and as such, may be suitable for development as a drug formulation in human clinical trials including patients with breast cancer.

## 4. Discussion

Curcumin-related studies have received great interest from the medicinal and scientific community owed to its multiple biological and therapeutic activities. Curcumin has been shown to control oxidative and inflammatory conditions, arthritis, metabolic syndrome, anxiety and hyperlipidemia [4]. Furthermore, it possesses antiviral, antimicrobial, immunomodulatory, antiangiogenic and anticancer activities [4]. It has been reported that curcumin may inhibit various signaling pathways associated with cancer including breast cancer [5,6]. Furthermore, several studies have revealed the inhibitory effect of curcumin on breast cancer [21,22]. Hu et al. demonstrated that curcumin inhibited the proliferation and induced the apoptosis of T47D, MCF7, MDA-MB-231 and MDA-MB-468 cells [21]. Another study demonstrated that curcumin induced apoptosis in breast cancer cells and inhibited breast tumor growth in mice [22]. Moreover, curcumin is known to target numerous signaling pathways associated with breast cancer, including p53, Ras, PI3K, protein kinase B, Wnt/β-catenin, mammalian target of rapamycin (mTOR), NF-κB, OPN, cyclooxygenase-2 (Cox-2) and MMP-9 [23,24,25]. Despite these therapeutic aspects, the efficacy of curcumin as an anticancer agent has been restricted by its poor solubility and bioavailability in various cancer types, including breast cancer [8]. Furthermore, researchers have classified curcumin under “pan-assay interference compounds” and “invalid metabolic panaceas” [20]. However, numerous preclinical and clinical studies with curcumin revealed a vast amount of data that reflect the therapeutic potential of it and suggest that the significant data generated in this field cannot be excluded without more attention. Nevertheless, various approaches have been investigated to improve the solubility and bioavailability of curcumin, among which the use of polymeric nanocarriers as a delivery vehicle is one of the most promising approaches. CHNPs are excellent candidates for curcumin delivery as they are biocompatible, biodegradable and mucoadhesive, and can be functionalized easily [10]. A range of methods for preparation allow researchers to customize synthesis according to their drug of interest. Several reports have shown that delivery through CHNPs can significantly enhance the therapeutic efficacy of curcumin in different cancer types, including breast cancer [16,17,18,19]. Anitha et al. demonstrated that curcumin-loaded N, O-carboxy methyl chitosan nanoparticles significantly inhibited breast cancer cell viability, while showing lower cytotoxicity towards normal cells [26]. Another report suggested that curcumin-loaded chitosan alginate nanospheres inhibited proliferation and increased apoptosis in breast cancer spheroid models [27]. Similarly, other studies have highlighted the therapeutic efficacy of curcumin–chitosan nanoformulations against breast cancer [28,29,30]. Despite this, Cur-CHNPs have not been successfully translated into clinical studies. The primary reason for this is a lack of comprehensive in vivo preclinical studies. To translate an experimental therapeutic into clinical studies, detailed preclinical assessment of antitumor efficacy, pharmacokinetics and toxicity are necessary using in vivo models. However, the majority of preclinical studies regarding curcumin-loaded chitosan nanocarriers are restricted to in vitro cellular uptake and cytotoxicity assessments. Hence, there is a need to examine the therapeutic efficacy and toxicity of Cur-CHNPs in preclinical in vivo models. In the present study, an extensive preclinical assessment was performed to examine the in vitro cellular uptake and cytotoxicity, as well as the in vivo biodistribution, tumor homing, therapeutic efficacy and toxicity of ionically crosslinked Cur-CHNPs in breast cancer.

Cur-CHNPs were synthesized by ionic gelation of positively charged chitosan by polyanionic TPP. This method does not involve the use of any organic solvents, thereby making CHNPs less toxic and more biocompatible. The various curcumin-loaded nanodelivery systems in breast cancer and other cancers are summarized in Table 2. The particle size and zeta potential of the formulation were analyzed by DLS. Small particle size has an extensive effect on the diagnostic and therapeutic efficacy of nanoparticles, with a large surface area, high drug loading capacity, biocompatibility and controlled drug release. Nanoparticles with a size < 200 nm retain a longer in-blood circulation and exhibit a higher accumulation in the tumor due to blood vessels in tumor vasculature having the highest pore size [31]. Hence, comprehensive optimization was performed in this study with respect to drug loading and NP size. We obtained stable nanoformulation with 2.5–25% curcumin to chitosan ratios. At 30 and 50% ratio, precipitates were observed which might be due to precipitation of curcumin at high concentration leading to interference in NP formation. Nanoformulation with 25% curcumin was chosen for further studies as it had highest amount of total drug loaded with a small increase in size as compared to other ratios (Table 1). The size of optimized Cur-CHNPs was found to be much lower than 150 nm, making them suitable for drug delivery. The encapsulation of curcumin in CHNPs was also confirmed by UV-visible and FTIR spectroscopy.

Furthermore, the long-term colloidal stability of Cur-CHNPs was examined and no significant agglomeration was observed over a period of 74 days suggesting the high stability of Cur-CHNPs. Colloidal stability is an important feature of nanoformulations as it affects the diffusive capability of nanoparticles in the tumor microenvironment [42]. Higher stability leads to enhanced diffusion and cellular internalization of nanoparticles in tumors, leading to an improved therapeutic efficacy. In the present study, in vitro cellular uptake experiments demonstrated the significant localization of CHNPs in breast cancer cells in a dose-dependent manner, depicting their efficient cell penetration capability, which may be attributed to the higher stability and positive zeta potential of CHNPs. Furthermore, Cur-CHNPs showed a higher inhibitory effect in MDA-MB-231 cells as compared to free curcumin. Moreover, Cur-CHNPs treatment in combination with 5-FU exhibited an enhanced inhibition of breast cancer cell viability compared with free curcumin or 5-FU alone. Several reports in the literature have suggested that doxorubicin and paclitaxel are being used in TNBC [43,44,45]. However, there are limited data available with the role of 5FU in inhibition of cell migration and downregulation of protumorigenic and angiogenic gene expression in breast cancer models. Because of that, we selected 5-FU alone or in combination with Cur-CHNPs in cell viability, migration and tumor-specific target gene expression in breast cancer models [46]. The effect of Cur-CHNPs with or without 5-FU was further examined in primary cells derived from breast cancer clinical specimens. Primary cells grown from patient tumor specimens are widely used in cancer research as they closely represent the tissue of origin. Since the cells are derived from tissue and are not modified, they maintain the genetic and pathophysiological properties of the tissue, making them a more suitable model for in vitro examination of therapeutic efficacy [43]. Similar to the MDA-MB-231 cells, Cur-CHNPs in the absence or presence of 5-FU exhibited an improved therapeutic efficacy against the breast cancer clinical specimen-derived primary cells as compared with free curcumin. These data suggested that Cur-CHNPs possessed a higher therapeutic efficacy than free curcumin and this effect may be replicated in clinical studies. Moreover, the use of Cur-CHNPs in combination with other anticancer agents may further enhance the therapeutic efficacy. The results of the wound-healing assay performed in the present study demonstrated that Cur-CHNPs and Cur-CHNPs plus 5-FU attenuated the motility of MDA-MB-231 cells compared with free curcumin, suggesting the antimigratory and potential antimetastatic properties of Cur-CHNPs and Cur-CHNPs plus 5-FU.

Several studies have demonstrated that the anticancer effectiveness of curcumin in triple-negative breast cancer is targeted via various important signaling pathways such as the PI3K/Akt/mTOR, JAK/STAT, Foxo, HIF-1, NF-κB, p53, Wnt/β-catenin, OPN, Cox-2 and MMP-9 pathways [23,24,25,46]. In the present study, to understand the function of Cur-CHNPs and Cur-CHNPs in combination with 5-FU on tumorigenesis and angiogenesis, MDA-MB-231 cells were treated with Cur-CHNPs plus 5-FU and the results revealed that this combination downregulated PI3K and Akt activation. In addition, Cur-CHNPs, either alone or in combination with 5-FU, reduced the expression levels of OPN and VEGF. These results suggested that Cur-CHNPs in combination with 5-FU inhibited breast cancer cell migration, PI3K and Akt activation, and OPN and VEGF expression in these cells.

To validate the in vitro findings of the present study using in vivo models, human orthotopic breast cancer models using NOD/SCID mice were developed. Efficient tumor targeting and biodistribution using in vivo conditions is necessary for nanocarriers to be employed as a drug delivery system in cancer therapy. In the present study, the in vivo imaging results revealed a high accumulation of CHNPs in the breast tumors following oral treatment, indicating their strong tumor-homing capability. It is well established that the vasculature of most solid tumors is impaired with gaps in the endothelial lining, enabling nanosized particles to easily accumulate in tumors compared with normal tissue. In addition, the lymphatic drainage system in tumors is defective, which does not allow quick clearance of nanoparticles from the tumor site. In combination, these phenomena led to tumor-specific accumulation and a higher retention of nanoparticles at the tumor site, which is termed the enhanced permeation and retention (EPR) effect. The EPR effect leads to passive targeting of tumors by the nanoformulation. Hence, due to the EPR effect, CHNPs are retained at the tumor site for a longer time compared with other organs, which might be the mechanism behind the notable tumor homing of CHNPs. Furthermore, the mucoadhesive properties of CHNPs improve their absorption through the gastrointestinal tract when administered orally. Due to this property, CHNPs interact with mucosa for a longer time, enhancing their absorption and reducing their clearance from the gastrointestinal tract. Furthermore, the mucoadhesive nature of nanoparticles enhances the contact time between the drug formulation and the oral mucosa leading to a higher bioavailability of the chemotherapeutic drug, and therefore therapeutic efficacy. In the present study, in addition to the tumors, the CHNPs were found to be significantly distributed in the liver and kidneys, which is expected owing to the higher blood supply in these organs. However, the CHNPs were mostly cleared from these organs within 72 h. In contrast, Cy5.5-CHNPs displayed a high tumor contrast for 72 h, demonstrating their long-term retention in tumors. These observations indicated the high tumor targeting ability of CHNPs and predicted their higher drug delivery potential and therapeutic efficacy in breast cancer. The in vivo antitumor efficacy results of the present study further validated these findings, as Cur-CHNPs treatment significantly reduced breast tumor growth compared with free curcumin in the orthotopic mice models. Therefore, these results suggested that CHNPs had an efficient tumor targeting potential and that curcumin delivery via CHNPs enhanced its therapeutic efficacy in breast cancer.

In the present study, the in vivo toxicity of Cur-CHNPs was further examined in healthy BALB/c mice. Nanotoxicity is an increasing concern for the translation of experimental nanocarriers into clinical therapeutics. Extensive research in the last decade has revealed various toxic effects of nanomaterials being used in different fields [47]. It has been reported that nanoparticles may induce multiple health concerns including inflammation, immunotoxicity, haematoxicity and genotoxicity, and may adversely affect the function of various organs such as the liver and kidneys [47]. Hence, preclinical toxicity evaluation of nanoformulations using in vitro and in vivo models is an important criterion for the translation of nanotherapeutics into clinics. Due to extensive blood supply, the liver and kidneys are preferential sites for nanoparticle accumulation in the body. Thus, evaluation of hemocompatibility and the effect on liver and kidney function is essential while examining nanotoxicity. In the present study, the acute and subchronic toxicity of Cur-CHNPs in healthy BALB/c mice were examined by assessing various parameters associated with hemotoxicity alongside liver and kidney function. The results revealed no significant alteration in these parameters in Cur-CHNP-treated mice compared with the untreated controls, in both the acute and subchronic assessments. As it is a biopolymer, chitosan possesses high biocompatibility and biodegradability, and has been approved by the US-FDA for wound-healing purposes. The LD_50_ of orally administered chitosan is as high as 16 g/kg of body weight [48]. These features make chitosan-based nanomaterials quite safe for drug delivery functions. Moreover, CHNPs synthesis by the ionic crosslinking method does not involve organic chemicals, further making it less toxic to biological tissue [23,49,50,51]. Previously, Hanafy et al. reported that nasal administration of galantamine/chitosan complex nanoparticles did not cause any toxicity or histopathological manifestations in rats [52]. Furthermore, reports have suggested that chitosan can be degraded in the body by enzymes such as lysozyme, chitinase and *N*-acetyl-β-D-glucosaminidase, indicating the biodegradability of chitosan [53,54,55]. Hence, the results of the present study demonstrated that Cur-CHNPs did not incur toxicity in healthy mice.

In conclusion, the present study provided a detailed characterization of the physicochemical, biological and toxicological properties of Cur-CHNPs. The results suggested that Cur-CHNPs had a high stability, efficient cellular uptake, strong tumor targeting capability and improved therapeutic efficacy, with no nonspecific toxicity. Further investigation of the therapeutic efficacy of Cur-CHNPs using patient-derived xenografts may be useful in the translation of these nanoformulations into clinics. In summary, the present study established that Cur-CHNPs possessed great potential to be translated into a potential druggable candidate, either alone or in combination with standard drugs, and may be translated into clinical trials against breast cancer.

## Figures and Tables

**Figure 1 nanomaterials-14-01294-f001:**
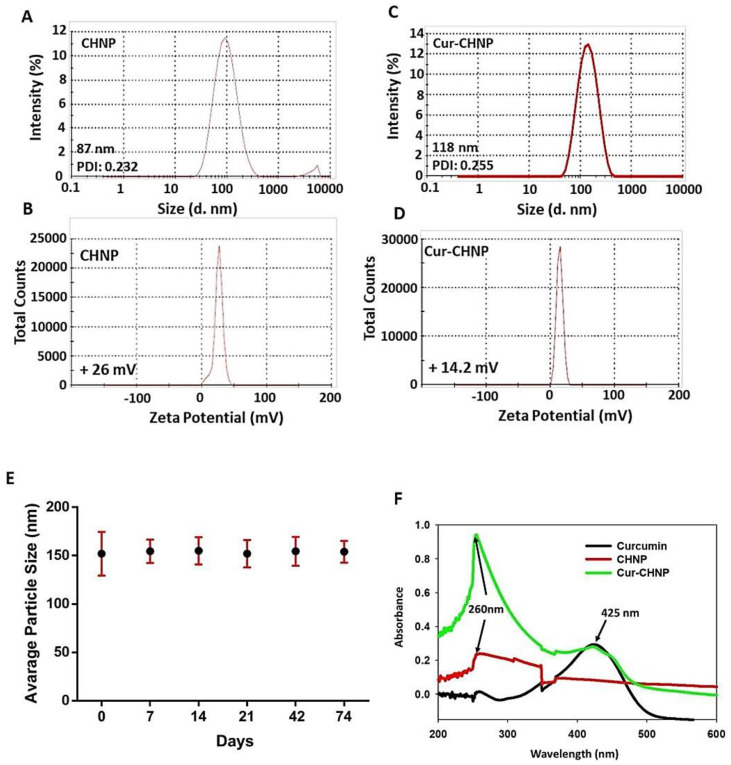
Physicochemical characterization of CHNPs and Cur-CHNPs. (**A**) Size and (**B**) zeta potential curves of CHNPs. (**C**) Size and (**D**) zeta potential curves of Cur-CHNPs. (**E**) Colloidal stability of Cur-CHNPs. The stability of Cur-CHNPs were examined by observing the size at different time points (0–74 days) (n = 3). (**F**) UV-visible spectra of curcumin, CHNPs and Cur-CHNPs. CHNPs, chitosan nanoparticles; Cur-CHNPs; curcumin-encapsulated CHNPs.

**Figure 2 nanomaterials-14-01294-f002:**
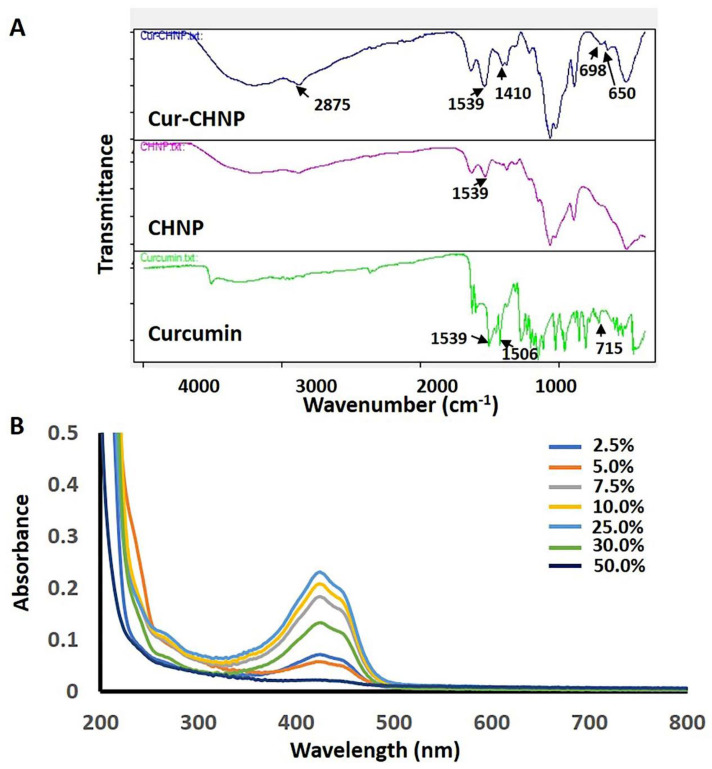
Drug encapsulation studies of Cur-CHNPs. (**A**) FTIR spectra of curcumin, CHNPs and Cur-CHNPs. (**B**) UV-visible spectra of Cur-CHNPs with different concentrations of curcumin (2.5, 5, 7.5, 10, 25, 30 and 50%, *w*/*w* of chitosan). CHNPs, chitosan nanoparticles; Cur-CHNPs; curcumin-encapsulated CHNPs; FTIR, Fourier transform infrared spectroscopy.

**Figure 3 nanomaterials-14-01294-f003:**
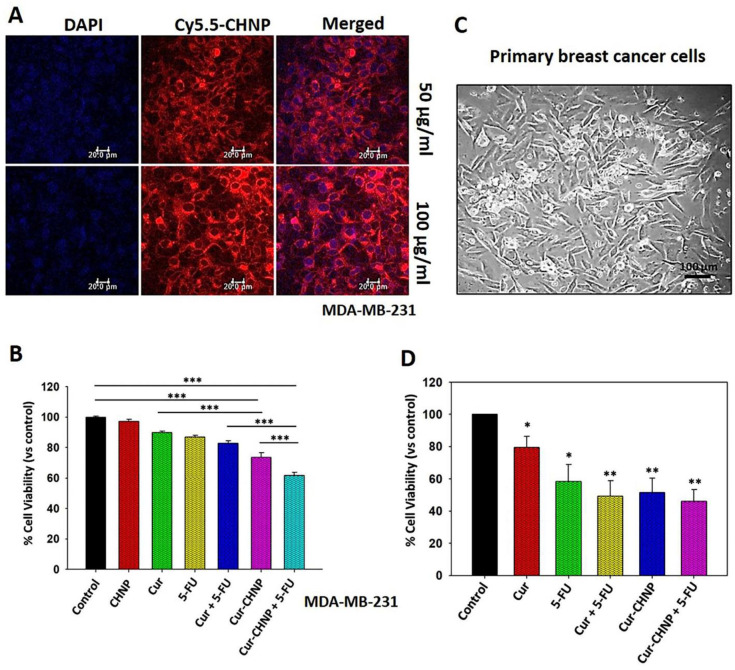
In vitro cellular uptake and antitumor efficacy of Cur-CHNPs in breast cancer cells. (**A**) In vitro cellular uptake of CHNPs was studied by fluorescence microscopy. Near-infrared fluorescent dye, Cy5.5 and conjugated CHNPs (50 and 100 µg/mL) were added to MDA-MB-231 cells for 2 h and images were analyzed by confocal microscopy. (**B**) MDA-MB-231 cells were treated with various combinations including free Cur or Cur-CHNPs, either alone or in combination with 5-FU, for 24 h and the MTT assay was performed (n = 4). (**C**) Phase contrast image (magnification, 10×) of the primary culture established from breast cancer clinical specimens. Scale bar, 100 µm. (**D**) Primary cells were treated with free Cur or Cur-CHNPs, either alone or in combination with 5-FU, for 24 h and the MTT assay was performed (n = 3). Error bars represent the mean ± SEM; * *p* < 0.05, ** *p* < 0.01, *** *p* < 0.001. 5-FU, 5-fluorouracil; CHNPs, chitosan nanoparticles; Cur, curcumin; Cur-CHNPs; curcumin-encapsulated CHNPs.

**Figure 4 nanomaterials-14-01294-f004:**
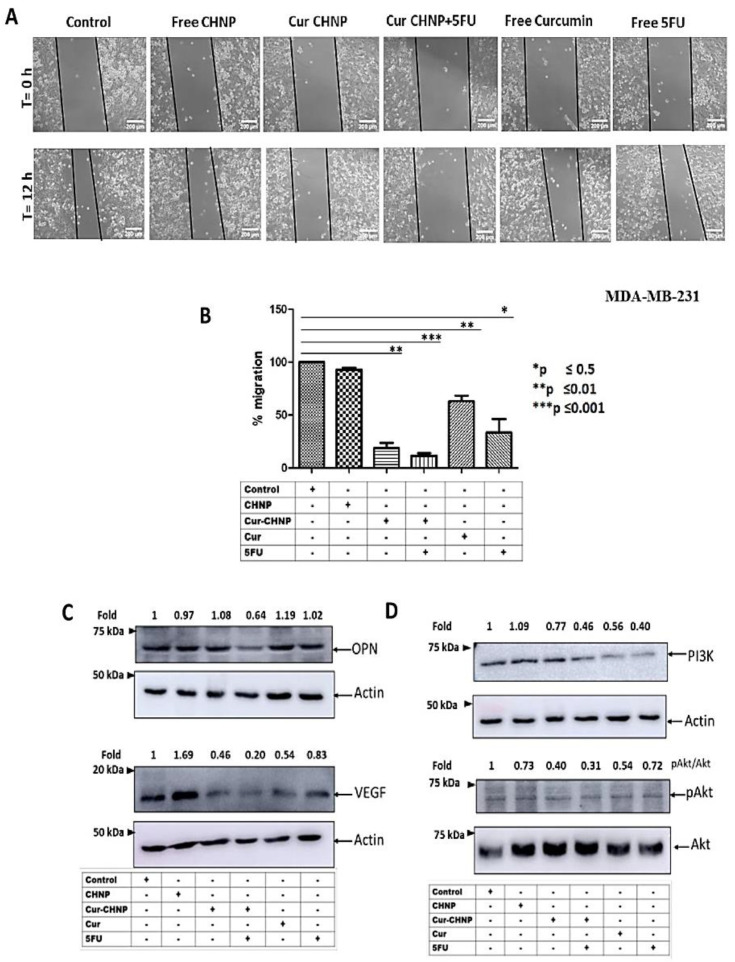
In vitro antitumor activity of Cur-CHNPs. MDA-MB-231 cells were seeded in a plate, wounded with a constant width, treated with CHNPs, Cur-CHNPs, Cur-CHNPs + 5-FU, curcumin and 5-FU and incubated at 37 °C for 12 h. (**A**) Images of the wounds were collected at 0 and 12 h. (**B**) The area migrated was analyzed using ImageJ software (Version 1.46r) and presented as a bar chart (n = 3). Effects of CHNPs, Cur-CHNPs, Cur-CHNPs + 5-FU, curcumin and 5-FU on (**C**) VEGF and OPN expression and (**D**) PI3K/pAkt activation in MDA-MB-231 cells were analyzed by Western blotting. Actin was used as the loading control. The error bars represent the mean ± SD; * *p* < 0.05, ** *p* < 0.01, *** *p* < 0.001. 5-FU, 5-fluorouracil; CHNPs, chitosan nanoparticles; Cur-CHNPs; curcumin-encapsulated CHNPs; OPN, osteopontin; VEGF, vascular endothelial growth factor.

**Figure 5 nanomaterials-14-01294-f005:**
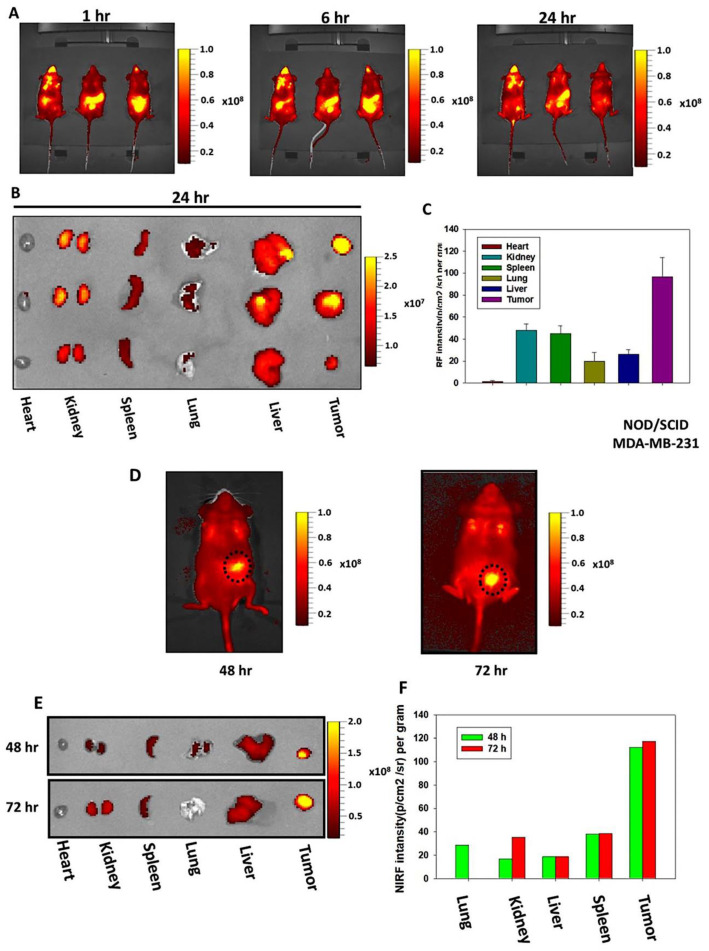
CHNPs exhibit efficient passive tumor targeting. (**A**) MDA-MB-231 cell-injected tumor-bearing NOD/SCID mice were treated with Cy5.5-CHNPs and NIRF images were captured at 1, 6 and 24 h (n = 3). (**B**,**C**) NIRF images of various major organs and tumors dissected from Cy5.5-CHNP-treated mice were collected under ex vivo conditions after 24 h. NIRF intensity was measured and plotted as a bar chart (mean ± SEM; n = 3). (**D**) Tumor-bearing mice were treated with Cy5.5-CHNP (5 mg/kg) and NIRF images were collected at 48 and 72 h (n = 1). (**E**,**F**) NIRF images of dissected major organs and tumors were collected by ex vivo imaging after 48 and 72 h. NIRF was measured and plotted as a bar chart (n = 1). CHNPs, chitosan nanoparticles; Cy5.5-CHNPs, Cy5.5 conjugated CHNPs; NIRF, near-infrared fluorescent.

**Figure 6 nanomaterials-14-01294-f006:**
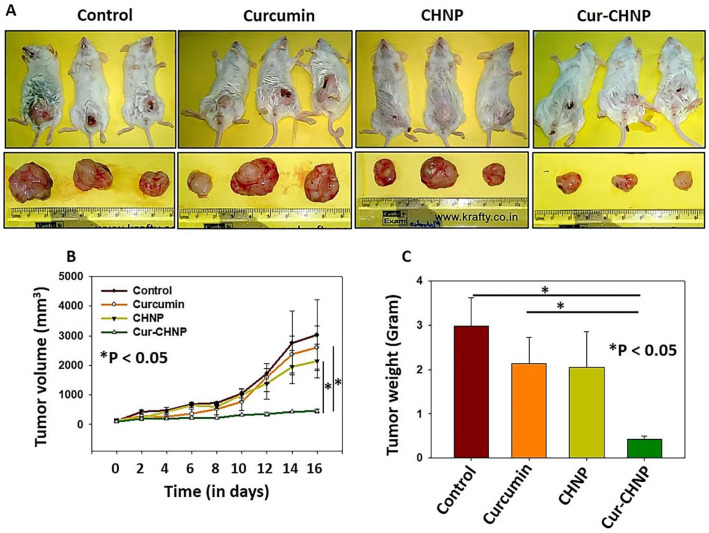
Cur-CHNPs significantly inhibit in vivo breast tumor growth. (**A**) Images of MDA-MB-231 cell-injected breast tumor-bearing mice and the excised tumors following various treatment conditions. (**B**) Tumor volumes were measured and statistically analyzed. The graph represents the change in the mean tumor volume with respect to time ± SEM. n = 3; * *p* < 0.05 vs. curcumin. (**C**) Tumors were weighed and analyzed statistically. Bar graph represents the mean tumor weight ± SD. n = 3; * *p* < 0.05. Cur-CHNPs; curcumin-encapsulated chitosan nanoparticles.

**Figure 7 nanomaterials-14-01294-f007:**
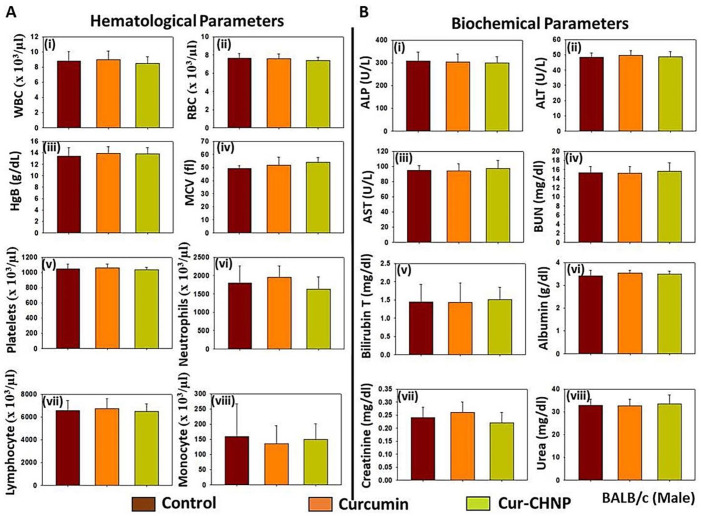
Assessment of the acute toxicity of Cur-CHNPs in BALB/c mice. Male BALB/c mice (4–6 weeks old) were treated orally with 20 mg/kg free curcumin or Cur-CHNPs (concentration corresponding to curcumin) every day for 1 week. After 1 week, blood was drawn through the retro-orbital plexus, transferred to potassium EDTA containing collection tubes and subjected to hematological and biochemical analysis. (**A**) Hematological parameters such as (i) WBC count, (ii) RBC count, (iii) Hb, (iv) MCV, (v) platelet count, (vi) neutrophil count, (vii) lymphocyte count and (viii) monocyte count were analyzed. Bars represent the mean ± SD (n = 4). (**B**) Biochemical parameters such as serum concentrations of (i) ALP, (ii) ALT, (iii) AST, (iv) BUN, (v) total bilirubin, (vi) albumin, (vii) creatinine and (viii) urea were analyzed as liver and kidney function makers. Bars represent the mean ± SD (n = 4). ALP, alkaline phosphatase; ALT, alanine aminotransferase; AST, aspartate aminotransferase; BUN, blood urea nitrogen; Cur-CHNPs; curcumin-encapsulated chitosan nanoparticles; Hb, hemoglobin; MCV, mean corpuscular volume; RBC, red blood cell; WBC, white blood cell.

**Figure 8 nanomaterials-14-01294-f008:**
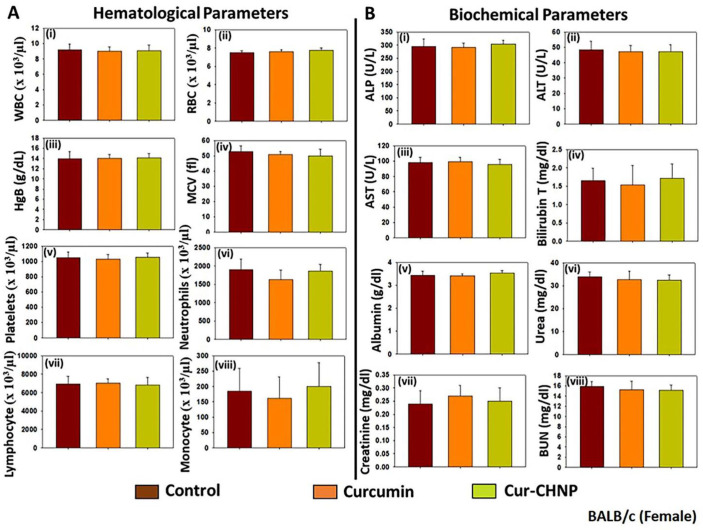
Assessment of the subchronic toxicity of Cur-CHNPs in female BALB/c mice. Female BALB/c mice (4–6 weeks old) were treated orally with 10 mg/kg free curcumin or Cur-CHNPs (concentration corresponding to curcumin) every alternate day for 28 days. Blood was drawn through the retro-orbital plexus into potassium EDTA containing collection tubes, and (**A**) hematological parameters such as (i) WBC count, (ii) RBC count, (iii) Hb, (iv) MCV, (v) platelet count, (vi) neutrophil count, (vii) lymphocyte count and (viii) monocyte count were analyzed. Bars represent the mean ± SD (n = 5). (**B**) Biochemical parameters such as serum concentrations of (i) ALP, (ii) ALT, (iii) AST, (iv) total bilirubin, (v) albumin, (vi) urea, (vii) creatinine and (viii) BUN were analyzed as liver and kidney function makers. Bars represent the mean ± SD (n = 5). ALP, alkaline phosphatase; ALT, alanine aminotransferase; AST, aspartate aminotransferase; BUN, blood urea nitrogen; Cur-CHNPs; curcumin-encapsulated chitosan nanoparticles; Hb, hemoglobin; MCV, mean corpuscular volume; RBC, red blood cell; WBC, white blood cell.

**Table 1 nanomaterials-14-01294-t001:** Encapsulation efficiency of curcumin-loaded CHNPs.

Curcumin (*w*/*w*%)	Particle Size (nm)	PDI	Appearance	EE %
**2.5**	92.96	0.334	Clear Suspension	63.14 ± 4.10
**5**	94.55	0.376	Clear Suspension	82.46 ± 5.28
**7.5**	93.14	0.312	Clear Suspension	53.62 ± 2.45
**10**	95.72	0.401	Clear Suspension	79.17 ± 2.23
**25**	106.5	0.348	Clear Suspension	77.14 ± 0.23
**30**	1364	1.0	Precipitate	90.28 ± 1.76
**50**	1315	1.0	Precipitate	94.42 ± 0.31

**Table 2 nanomaterials-14-01294-t002:** Curcumin nanodelivery systems in various cancers.

Nature of Curcumin Nanoparticles	Cancer Types	Specific Molecular Target	References
MSN-HA-C (Hyaluronic acid functionalized mesoporous silica nanoparticles loaded with curcumin)	Breast Cancer	NF-κB and Bax	[32]
Peptide-HSA/Cur NPs (PDL1 binding peptide conjugated–Human serum albumin–curcumin nanoparticles)	Breast Cancer	PDL1, Apoptosis	[33]
CaCO_3_@Cur@QTX125@HA (CaCO_3_ nanoparticles loaded with curcumin (Cur) and Histone Deacetylase (HDAC) inhibitor, QTX125, and coated with hyaluronic acid)	Colorectal cancer	Apoptosis	[34]
HSA-Curcumin NPs (Human serum albumin-conjugated curcumin NPs)	Breast cancer	Apoptosis	[35]
Tf-CRC-SLNs (Curcumin-loaded transferrin-bioconjugated solid lipid nanoparticles)	Prostate cancer	Apoptosis	[36]
PEG-FA@Nio-Cur (Folate-targeted curcumin-loaded niosomes)	Breast cancer	Bax, Bcl2, p53	[37]
Cur@ZIF-8@HA (Hyaluronic acid-coated curcumin-loaded ZIF-8 nanoparticles)	Breast cancer	Apoptosis and induction of ROS	[38]
Cur-NPs (Curcumin-loaded PLGA nanoparticles)	Gastric cancer	Cell cycle arrest and Apoptosis	[39]
PGMD-Cur nanoparticles (poly-glycerol–malic acid–dodecanedioic acid)/curcumin nanoparticles)	Breast cancer	Caspase 9	[40]
Cur.SA-loaded CM nanoparticles (Succinylated Cur-encapsulated mannosylated-chitosan nanoparticles)	Colon cancer	PARP, Caspase 8	[41]

## Data Availability

The data generated in the present study may be requested from the corresponding author.

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
