# Peer review of "Chitosan Nanoparticle-Mediated Delivery of Curcumin Suppresses Tumor Growth in Breast Cancer"

_nanomaterials, 2024, doi:10.3390/nano14151294_

Round 1
Reviewer 1 Report
Comments and Suggestions for Authors
In this manuscript, the authors showed the development and characterization of a curcumin-loaded chitosan nanoparticle. In addition, The authors presented a robust biological evaluation of these nanoparticles with interesting results.
General comments
1- The approval number of all ethics committee protocols (human and animal) must be provided.
2- The quality of all images must be improved.
Specific Comments
1- Section 3.3: A minimal characterization of the primary human breast cancer cell must be presented for a valid comparison with MDA-MB-231 cell line in terms of antitumor activity.
2- Section 3.9: As described in the in vitro activity, the combinations of 5-FU and curcumin nanoparticles significantly reduced the cell viability. Does this combination enhance systemic toxicity? I suggest including at least the toxicity of 5-FU alone and the combination with the nanoparticle in the acute toxicity assay.
Author Response
General comments
Question 1: The approval number of all ethics committee protocols (human and animal) must be provided.
Response: We thank the reviewer for their comments. We have included the approval numbers of all ethics committee protocols (human and animal) in the revised manuscript.
2- The quality of all images must be improved.
Response: We thank the reviewer for their keen observation. We have improved the quality of the images to 300 dpi.
Specific Comments
1- Section 3.3: A minimal characterization of the primary human breast cancer cell must be presented for a valid comparison with MDA-MB-231 cell line in terms of antitumor activity.
Response: We extend our gratitude for the insightful comments. We have used the MDA-MB-231 cell line as control as we had limited patient samples and few primary cultures were successfully established. However we have included both the primary human breast cancer cells as well as MDA-MB-231 cell line in terms of cell viability assays for checking the efficiency of nanoformulation in these cultured cells. Earlier we have established and characterised many primary cultures from breast cancer patient derived xenograft (PDX) models and recently published (Butti et al, Oncol Rep. 2023 May; 49(5): 99. doi: 10.3892/or.2023.8536. Epub 2023 Mar 31.PMID: 36999625).
2- Section 3.9: As described in the in vitro activity, the combinations of 5-FU and curcumin nanoparticles significantly reduced the cell viability. Does this combination enhance systemic toxicity? I suggest including at least the toxicity of 5-FU alone and the combination with the nanoparticle in the acute toxicity assay.
Response: We are thankful to the learned reviewer for this insightful suggestion.
However, due to lack of time and resources we won’t be able to perform this experiment. Moreover, the focus of this study was preparation of a biocompatible curcumin nanoformulation with improved therapeutic efficacy against breast cancer. Hence, we examined the systemic toxicity of Cur-CHNP formulation in this study. The combination with 5-FU was an additional study. It will require a separate comprehensive study to assess the therapeutic efficacy and systemic toxicity of Cur-CHNP in combination with other anti-cancer drugs including 5-FU.
Reviewer 2 Report
Comments and Suggestions for Authors
The manuscript by Mishra, Yadav, Malhotra and co-workers proposed the encapsulation of curcumin as therapeutic agents in chitosan nanoparticle for the treatment of breast cancer.
This reviewer thinks that this manuscript could be publishable in Nanomaterials. However, it would be great if the authors take into consideration the following comments/additions:
1. In the abstract, the authors describe that "... the therapeutic impact of curcumin against cancer, especially breast cancer, has been constrained...". Is there any evidence that curcumin has been limited in breast cancer compared to other cancers? I suggest that the authors clarify this.
2. In line 229, the authors discuss that the size and Z-potential of Cur-CHNP is 118 nm and +14.2 mV, respectively. But in Table 1, any nanoparticles presented have those characteristic. Please the author needs to further discuss the loading differences of these nanoparticles with the data presented in Table 1.
3. In Table 1, I suggest adding the Z-potential data for each nanoparticle included in it.
4. In Figure 3D, it would be necessary to add the result of testing CHNP in primary cells as a control.
5. I suggest adding a summary table that includes studies using curcumin encapsulated in nanoparticles to treat cancer. This would be useful for people interested in the topic.
6. This reviewer disagrees with the title of the manuscript because the authors do not demonstrate that the nanoparticles developed are specifically "targeted" delivery. I suggest changing it.
7. I suggest performing in vitro experiments with others breast cancer cell lines, including healthy cells, to verify the efficacy of the nanosystem.
8. I suggest performing in vitro experiments using Cur-CHNP where both curcumin and 5FU are encapsulated. It does not make sense to use an encapsulation system and then a free anticancer drug to enhance the treatment. If 5FU cannot be encapsulated, test doxorubicin as an anti-cancer drug to treat breast cancer.
9. The authors have no results with Cur-CHNP and 5FU in mice to complete Fig. 6? Since this was the most effective system in vitro, why didn't the authors follow up with in vivo studies? Please discuss this in the manuscript.
10. I suggest adding up-to-date references between 2023-2024 on the topic.
Author Response
- In the abstract, the authors describe that "... the therapeutic impact of curcumin against cancer, especially breast cancer, has been constrained...". Is there any evidence that curcumin has been limited in breast cancer compared to other cancers? I suggest that the authors clarify this.
Response: We are thankful to the reviewer for comments. We have modified the sentence as, “However, due to its poor water solubility and bioavailability, the therapeutic impact of curcumin against cancer, including breast cancer, has been constrained”.
- In line 229, the authors discuss that the size and Z-potential of Cur-CHNP is 118 nm and +14.2 mV, respectively. But in Table 1, any nanoparticles presented have those characteristic. Please the author needs to further discuss the loading differences of these nanoparticles with the data presented in Table 1.
Response: We are thankful to the reviewer for their keen observation. However, the data presented in Table 1 demonstrate the optimization of nanoformulation with respect to drug loading and NP size. The data represented in the Figure 1 C &D is representative image of one of multiple size and zeta potential measurements after optimized nanoformulation. As per reviewer’s suggestion, we have discussed it further (Discussion part) in the revised manuscript.
- In Table 1, I suggest adding the Z-potential data for each nanoparticle included in it.
Response: We are thankful to the reviewer for valuable suggestion. Since we have not checked for the Z-potential of Cur-CHNPs for all the concentrations of curcumin (2.5% -50%). The stable concentration of 25% of curcumin was checked for the characterizations as it was used for all the experimental studies in the manuscript. Therefore we could not include the Z-potential for all the concentrations of curcumin in the study.
- In Figure 3D, it would be necessary to add the result of testing CHNP in primary cells as a control.
Response: We are thankful to the reviewer for the insightful comments. It was difficult to establish the primary culture and perform this assay in the given time frame. Earlier we have established and characterised many primary cultures from breast cancer patient derived xenograft (PDX) models and recently published (Butti et al, Oncol Rep. 2023 May; 49(5): 99. doi: 10.3892/or.2023.8536. Epub 2023 Mar 31.PMID: 36999625).
- I suggest adding a summary table that includes studies using curcumin encapsulated in nanoparticles to treat cancer. This would be useful for people interested in the topic.
Response: We are thankful to the reviewer for the invaluable suggestion. As per the reviewer suggestion we have included a summary table (Table no 2) under the discussion section.
- This reviewer disagrees with the title of the manuscript because the authors do not demonstrate that the nanoparticles developed are specifically "targeted" delivery. I suggest changing it.
Response: We have changed the title of our manuscript as “Chitosan Nanoparticle-mediated Delivery of Curcumin Suppresses Tumor Growth in Breast Cancer”.
- I suggest performing in vitro experiments with others breast cancer cell lines, including healthy cells, to verify the efficacy of the nanosystem.
Response: We are thankful to reviewer for the critical insights. Although we have not investigated the cytotoxicity of nanocarriers in healthy cells, we have evaluated the in vivo toxicity of Cur-CHNPs in healthy Balb/c male and female mice.
- I suggest performing in vitro experiments using Cur-CHNP where both curcumin and 5FU are encapsulated. It does not make sense to use an encapsulation system and then a free anticancer drug to enhance the treatment. If 5FU cannot be encapsulated, test doxorubicin as an anti-cancer drug to treat breast cancer.
Response: We admire reviewer’s critical suggestion. There are many reports showing synergistic effect of free curcumin with other anti-cancer drugs including 5-FU against cancer. Here, we aimed to examine whether delivering curcumin with chitosan nanoparticles can enhance this effect or not. We also tried to encapsulate 5-FU with curcumin in chitosan nanoparticles but the formulation was unstable with large size and ineffective loading. Hence, we dropped the idea of 5-FU encapsulation. Moreover, the major focus of this study was preparation of a biocompatible curcumin nanoformulation with improved therapeutic efficacy against breast cancer. The combination with 5-FU was an additional study.
- The authors have no results with Cur-CHNP and 5FU in mice to complete Fig. 6? Since this was the most effective system in vitro, why didn't the authors follow up with in vivo studies? Please discuss this in the manuscript.
Response: As discussed in previous response, this study was focused on preparation of a biocompatible curcumin nanoformulation with improved therapeutic efficacy against breast cancer. The combinatorial treatment was a supplementary study. It will require a separate comprehensive in vitro and in vivo study to assess the therapeutic efficacy and systemic toxicity of Cur-CHNPs in combination with other anti-cancer drugs including 5-FU.
- I suggest adding up-to-date references between 2023-2024 on the topic.
Response: We have added new up-to-date references in our revised manuscript as per reviewers’ suggestion.
I hope now the revised manuscript, after incorporating the modifications and answering all the questions raised by the reviewers, will be suitable for publication in Nanomaterials.
Round 2
Reviewer 1 Report
Comments and Suggestions for Authors
The authors successfully clarified all my questions.